# Refractory Multiple Myeloma in a West Highland White Terrier: Clinical Presentations and Therapeutic Interventions

**DOI:** 10.3390/ani15162405

**Published:** 2025-08-16

**Authors:** Hyomi Jang, Hyejin Jeong, A Sa Sung, Hyojun Kwon, Jiheui Sohn, Jong-In Kim, Moon-Yeong Choi, Chan Huh, Dong-In Jung

**Affiliations:** 1VIP Animal Medical Center (Cheongdam), Seoul 06068, Republic of Korea; hm100su@gmail.com (H.J.); hjin5160@gmail.com (H.J.); aniki1017@naver.com (A.S.S.); hyojun1220@naver.com (H.K.); wlgml924@naver.com (J.S.); jjongin88@vipah.co.kr (J.-I.K.); 2S Animal Cancer Center, Yangsan 50608, Republic of Korea; vetmoon7@gmail.com (M.-Y.C.); 24sacc@naver.com (C.H.); 3College of Veterinary Medicine, Gyeongsang National University, Jinju 52828, Republic of Korea

**Keywords:** dog, radiation therapy, tyrosine kinase inhibitor, proteasome inhibitor

## Abstract

Multiple myeloma, a rare cancer in dogs, is characterized by abnormal plasma cell proliferation and immunoglobulin overproduction. This case report describes a West Highland White Terrier with relapsed and refractory multiple myeloma treated with various therapies, including conventional chemotherapy, radiation, and newer drugs, such as verdinexor and bortezomib. Verdinexor showed the longest stable remission with fewer side effects, while bortezomib was effective, but limited by its toxicity. Despite aggressive treatment, the dog eventually died from hyperviscosity syndrome complications. This report highlights the potential of novel therapies and the importance of supportive care for advanced canine multiple myeloma.

## 1. Introduction

Multiple myeloma (MM) is a malignant neoplasm characterized by clonal proliferation of plasma cells or their precursors, which produce excessive amounts of immunoglobulins. These neoplastic cells typically accumulate within the bone marrow, leading to a wide array of systemic and localized clinical manifestations. In dogs, MM is considered relatively uncommon, accounting for <1% of all malignant neoplasms, <10% of malignant hematopoietic tumors, and approximately 3.6% of primary and secondary bone tumors [1,2,3,4,5,6].

Definitive diagnosis of MM generally relies on histopathological examination of neoplastic tissue acquired through bone marrow aspiration or core biopsy [5]. However, in cases where biopsy is not feasible or histological findings are inconclusive, a presumptive diagnosis can be made if at least two of the following criteria are met: presence of monoclonal or biclonal gammopathy due to excessive production of clonal immunoglobulin paraproteins (M-proteins), detection of free light chains (Bence–Jones proteins) in the urine, presence of osteolytic lesions, or infiltration of plasma cells comprising over 20% of bone marrow cellularity [5,7].

Common clinical signs observed in affected dogs include lethargy, weakness, and decreased appetite, although clinical presentation may vary depending on the tumor location [5,7]. Bone marrow involvement can result in anemia, thrombocytopenia, and, less commonly, neutropenia. Osteolysis and bone invasion may lead to pain, pathological fractures, or neurological deficits if the spine is involved [5]. Additionally, secondary complications, such as hyperviscosity syndrome (HVS) due to hyperglobulinemia, paraneoplastic hypercalcemia, and coagulopathy caused by the interference of M-components with platelet aggregation may occur [5,6,7,8].

The classical first-line treatment for canine MM is a combination of melphalan and prednisolone (MP protocol), which has demonstrated an overall response rate ranging from 76% to 94%, depending on the protocol used [7,9]. The reported median overall survival time is approximately 930 days (range, 70–1554 days) [7]. Dogs unresponsive to this regimen or with uncontrolled disease often receive rescue therapy with other alkylating agents such as cyclophosphamide, chlorambucil, lomustine, vincristine, or doxorubicin [5,7,10,11]. In cases where solitary extramedullary plasmacytomas are identified, local treatment with radiation therapy (RT) may be beneficial [12,13,14,15].

Recent advances in veterinary oncology have introduced novel anti-cancer agents such as rabacfosadine—an acyclic nucleotide phosphonate, and verdinexor—a selective inhibitor of nuclear export. These drugs have demonstrated efficacy in the treatment of B-cell lymphoma and may offer therapeutic potential in MM due to their reduced systemic toxicity compared to conventional alkylating agents [7,16,17,18,19]. In human medicine, therapeutic protocols incorporating thalidomide, bortezomib, and lenalidomide are commonly employed as first-line treatments or for managing relapsed disease. Several drugs with similar mechanisms of action have been investigated in veterinary oncology, including a recent case report describing the successful use of bortezomib in a cat with MM [20,21,22,23,24]. Additionally, protein kinase inhibitors are under active investigation in human MM therapy and may represent a future direction for veterinary treatment strategies [25,26].

With timely intervention and adequate supportive care, canine MM typically carries a favorable short-term prognosis. However, the presence of hypercalcemia, extensive bone lysis, or Bence–Jones proteinuria is associated with poorer outcomes [6].

This report describes a case of relapsed and refractory MM in a West Highland White Terrier, managed with multiple therapeutic agents not previously reported in veterinary MM treatment. This case report highlights the clinical course, diagnostic imaging findings, therapeutic responses, and adverse effects associated with each treatment strategy.

## 2. Case Description

A 7-year-old neutered male West Highland White Terrier (body weight, 8.95 kg) was referred to a local animal hospital with a primary complaint of bilateral hindlimb ataxia. Initially, the clinical signs resembled lameness, and joint disease was suspected. A non-steroidal anti-inflammatory drug (NSAID) was administered for 3 weeks; however, the clinical signs progressively worsened. Subsequently, computed tomography (CT) and magnetic resonance imaging (MRI) were performed.

Imaging revealed a mass in the third lumbar vertebral body compressing the spinal cord at the L3–L4 level. Additional masses were detected in multiple vertebral bodies, including the ilium, pubis, and ribs. Enlarged mediastinal and abdominal lymph nodes, hepatomegaly with hypoattenuated hepatic lesions, and contrast-enhanced parenchymal changes in the liver, spleen, right adrenal gland, and kidneys (suggestive of renal infarction) were also observed, raising suspicion for MM (Figure 1 and Figure 2). The dog was then referred to our hospital for chemotherapy.

On physical examination, the patient’s systolic blood pressure was 160 mmHg, indicating mild hypertension. Hematologic evaluation revealed mild anemia (hematocrit, 30.8%) with a visually dark-red, viscous appearance, along with marked hyperproteinemia (total protein, 19.6 g/dL), hyperglobulinemia (17.4 g/dL), and mild hypercalcemia (12.9 mg/dL) (Table 1). Hematological smears stained with Diff-Quik revealed agglutinated and clumped red blood cells and numerous ghost cells on a dark blue background.

Further evaluation for MM included serum protein electrophoresis, which demonstrated monoclonal gammopathy in the β2 region, and urinalysis, which revealed positive Bence–Jones proteinuria (IDEXX Laboratories, Westbrook, ME) (Figure 3).

Fine-needle aspirations (FNAs) of the spleen, liver, and peripheral lymph nodes were performed. Cytological analyses of the spleen and liver revealed numerous plasma cells, characterized by round nuclei, basophilic cytoplasm, and prominent perinuclear clear zones (Figure 4). No significant neoplastic cells were observed in the peripheral lymph nodes.

Based on imaging and cytologic findings, a presumptive diagnosis of MM involving the vertebrae, ilium, pubis, ribs, and abdominal organs (liver, spleen, and thoracoabdominal lymph nodes) was made. Due to the patient’s poor clinical condition, histopathology and bone marrow biopsy were not performed.

Given the severe systemic compromise and circulatory disturbance likely secondary to hyperviscosity, supportive hospitalization with intravenous fluids and analgesics was initiated. Melphalan (0.1 mg/kg PO q24h) and prednisolone (0.5 mg/kg PO q24h) were prescribed as first-line therapies for MM. Along with chemotherapy, periodic blood tests, thoracic and abdominal radiographs, and abdominal ultrasonography were performed to monitor the progression and characteristics of the lesions over time.

Nine days later, the total protein and globulin concentrations had decreased to 16.3 and 13.9 g/dL, respectively. However, the patient remained in poor general condition, with persistent circulatory issues from blood hyperviscosity, marked abdominal pain, and significantly elevated canine pancreatic lipase (1828.3 µg/L), consistent with pancreatitis (Table 1).

Consequently, melphalan and prednisolone were discontinued, and a vincristine-doxorubicin-dexamethasone (VAD) chemotherapy protocol was initiated to induce a rapid reduction in blood protein concentration: vincristine (0.7 mg/m^2^ IV on days 8 and 15), doxorubicin (30 mg/m^2^ IV every 21 days), and dexamethasone (1 mg/kg IV on days 1, 8, and 15).

On day 20 (8 days after VAD initiation), the serum total protein and globulin levels further declined to 13.6 and 9.2 g/dL, respectively. However, by day 24, the patient exhibited acute loss of hindlimb proprioception, perineal reflex, and voluntary urination. Palliative-intent RT was planned to alleviate spinal compression from the L3–L4 vertebral mass.

For treatment planning, CT simulation was performed, and pre-and post-contrast images were imported into the radiation treatment planning system (Monaco, Elekta AB, Stockholm, Sweden). Volumetric modulated arc therapy (VMAT) was planned using a 6 MV linear accelerator (Elekta Synergy). Gross tumor volume (GTV) included the enhancing lesion at L3–L4 and osteolytic changes from T11 to L6. Clinical target volume (CTV) was defined as a 3 mm isotropic margin surrounding the GTV. A total dose of 20 Gy was delivered in five fractions (4 Gy/day over 1 week).

RT was administered concurrently with the VAD protocol for approximately 2 months. Two days after completing radiotherapy, the patient’s hindlimb paralysis and urinary dysfunction resolved. On day 53, serum globulin levels increased, prompting administration of cyclophosphamide (1 mg/kg PO q24h) for 7 days, leading to a reduction in globulin levels. By day 81, total protein and globulin had decreased to 6.5 and 3.5 g/dL, respectively (Table 1). However, hematocrit and platelet counts declined to 19% and 54 K/μL, respectively, indicating bone marrow suppression. The patient also exhibited vomiting, anorexia, and lethargy, prompting a reassessment of the disease status with CT imaging.

Follow-up CT showed significant regression of the vertebral mass and resolution of the spinal cord compression. The mass exhibited fat attenuation (ࢤ88 HU) and mild contrast enhancement (ࢤ46 HU) (Figure 5). Mediastinal lymph nodes were also reduced. However, osteolytic lesions in the vertebrae, ilium, and pubis persisted, and enlargement of abdominal lymph nodes and adrenal glands, as well as progression of renal infarction, were noted.

Despite VAD-related side effects, the patient remained clinically stable, allowing for the transition to verdinexor (1.25 mg/kg PO twice weekly, increased to 1.5 mg/kg), which was administered for maintenance. Over the following weeks, regular bloodwork, thoracic radiographs, and abdominal ultrasound showed stable mass size and serum protein levels (Table 1), with preserved body condition and weight.

On day 190, after approximately 3 months of verdinexor therapy, the patient presented with elevated serum protein levels, severe gastrointestinal signs, and lethargy (Table 1). Abdominal ultrasonography revealed a large intra-abdominal mass (54.2 × 49.1 mm), presumed to be hepatic lymphadenopathy, a duodenal papillary mass (8.1 × 20.2 mm), and a hypoechoic mass (11.8 × 8.7 mm) at the pancreatic body, suggesting metastasis (Figure 6). Combination therapy with verdinexor (1.67 mg/kg PO twice weekly), doxorubicin (30 mg/m^2^ IV), and dexamethasone (1 mg/kg IV) was initiated, followed by melphalan (7 mg/m^2^ PO q24h for 5 days). Although globulin levels decreased post-treatment, severe bone marrow suppression (non-regenerative anemia, thrombocytopenia, and leukopenia) and persistent clinical decline necessitated the discontinuation of all agents.

Following the recurrence of hyperglobulinemia, verdinexor and lomustine (70 mg/m^2^ PO once) were administered together but failed to produce meaningful clinical or laboratory improvements. On day 224, verdinexor was discontinued, and bortezomib (1.3 mg/m^2^ IV twice weekly) and dexamethasone (1 mg/kg IV weekly) were administered. Bortezomib was dose-escalated over 3 weeks to 1.65 mg/m^2^. On day 238, after 14 days of therapy, the patient showed clinical improvement and decreased total protein and globulin levels (Table 1). Abdominal ultrasonography revealed a dramatic reduction in previously detected masses, including hepatic and pancreatic lesions.

Bortezomib and dexamethasone were administered for 68 days. Apart from mild thrombocytopenia and gastrointestinal symptoms, no severe adverse effects were observed. The minimal effective bortezomib dose was determined to be 1.45–1.65 mg/m^2^ IV twice weekly, administered alongside melphalan or verdinexor with dexamethasone.

Subsequent discontinuation of verdinexor was required due to lethargy, nausea, fever, diarrhea, and worsening cytopenia. Although bortezomib alone was continued, hyperproteinemia and hyperglobulinemia progressed. Toceranib (3 mg/kg PO, thrice weekly) was added, but no clinical improvement was observed. On day 296, bortezomib was replaced with carfilzomib (70 mg/m^2^ IV once weekly), which preserved clinical stability but did not improve laboratory values.

On day 315, the patient presented with vomiting, diarrhea, anorexia, anemia, hyperproteinemia, hyperglobulinemia, and hypercalcemia (Table 1). Worsening hepatic lymphadenopathy was noted on ultrasound, prompting reinitiation of bortezomib (1.65 mg/m^2^ IV twice weekly). A lack of clinical response led to the administration of ixazomib (1 mg/dog PO weekly) with dexamethasone. However, after 3 weeks, hyperglobulinemia persisted and anemia worsened, prompting discontinuation. Emergency phlebotomy and red blood cell transfusion were attempted but proved ineffective.

Sorafenib (3 mg/kg PO q24h) was initiated on day 339; however, no significant improvement was observed after 20 days, even with bortezomib co-administration (Table 1). Combined sorafenib and bortezomib treatment was continued for 33 days. Although a transient globulin reduction was observed, rebound hyperglobulinemia, anemia, thrombocytopenia, and dermatological signs (alopecia, hyperpigmentation, and eruption at multiple sites) developed (Table 1, Table 2 and Figure 7). Urinalysis revealed hematuria associated with concurrent urinary tract infection, and hematochezia due to colitis was also observed (Table 2).

MRI was performed on day 394 to investigate newly developed urinary retention. No central nervous system lesion was identified. The L3 vertebral lesion exhibited a fat signal on both T1- and T2-weighted images and was reduced in size with minimal spinal cord compression (Figure 8).

Despite supportive care for urinary dysfunction, pain control, antibiotics, and blood transfusion, the patient developed a fever and bloody diarrhea. The dog died on day 398. Table 1, Figure 9, and Table 2 present the patient’s hematologic progression and therapeutic summary.

## 3. Discussion

This report describes a canine case of MM managed with a combination of conventional chemotherapeutic agents, including melphalan, vincristine, doxorubicin, and glucocorticoids, as well as novel modalities such as radiation therapy (RT), tyrosine kinase inhibitors (TKIs), selective inhibitors of nuclear export (SINEs), and proteasome inhibitors. During an approximate one-year treatment period, the patient received both inpatient and outpatient care, alongside a variety of supportive interventions aimed at managing tumor-related clinical signs and mitigating chemotherapy-associated adverse effects. These included supportive nutritional therapy, analgesics, antioxidants, antiemetics, antidiarrheals, appetite stimulants, and intravenous fluid administration.

In canine MM, the first-line chemotherapy generally consists of an alkylating agent, most commonly melphalan, combined with prednisolone [5,7,9,11]. In this case, low-dose daily melphalan therapy resulted in a decrease in serum globulin from 17.4 to 13.9 g/dL over 9 days, indicating antitumor efficacy. However, treatment was discontinued due to the deterioration in the patient’s overall condition, likely secondary to pancreatitis associated with hyperviscosity syndrome (HVS).

A previous study compared two melphalan-based protocols: the daily dose MP protocol (0.1 mg/kg/day for 10 days followed by 0.05 mg/kg/day maintenance) and the pulse-dose MP protocol, which used a higher melphalan dose (7 mg/m^2^/day—approximately 3.5 times higher than the daily dose based on the patient’s weight) for 5 days followed by a 3-week drug-free interval. Although the overall response rates (ORRs) were 79% and 94%, respectively, no statistically significant differences in efficacy or adverse events were observed [7]. Notably, several patients initially administered with the daily dose protocol achieved stable disease or complete remission after switching to the pulsed-dose regimen. The authors also reported that combining 1–2 doses of cyclophosphamide within 10 days of starting melphalan accelerated remission, suggesting that early intensification or combination with other alkylating agents may improve outcomes.

Another study applying a 10-day cycle MP protocol using low-dose melphalan (mean 2 mg/m^2^/day, equivalent to 0.1 mg/kg/day in a 9 kg dog) showed that 59% of 17 dogs achieved complete remission (CR), and 76% had a measurable response [9]. However, the median time to maximal globulin reduction was approximately 3 months (range, 15–444 days), indicating that this approach may be inadequate for managing acute and severe hyperproteinemia or HVS, as seen in the present case. Therefore, a pulse-dose MP protocol or early treatment escalation (e.g., switching protocols or combining with other alkylating agents) was implemented, and earlier control of the tumor burden and HVS might have been achieved. Nevertheless, due to the rapid clinical decline from pancreatitis and abdominal pain, a transition to a more aggressive protocol with vincristine, doxorubicin, and dexamethasone (VAD) was warranted, resulting in a prompt reduction in serum protein levels.

The VAD protocol, historically used as a primary or salvage treatment for human MM [28], is recommended in veterinary medicine primarily for relapsed or refractory MM following alkylating agent failure [11]. However, clinical reports of its veterinary applications are limited. In one case, a dog initially treated with cyclophosphamide and daily MP achieved partial remission but later relapsed and was treated with VAD for 2 months, maintaining normal globulin levels [29]. Unfortunately, treatment discontinuation for non-medical reasons led to tumor progression, and subsequent reinitiation of VAD failed to induce remission. Additionally, the VAD protocol is associated with severe bone marrow suppression and gastrointestinal side effects, requiring dose adjustments for doxorubicin. In contrast to the protocol for low-dose doxorubicin (6.5–8 mg/m^2^/day for 4 consecutive days), this case followed a conventional high-dose schedule (30 mg/m^2^ every 3 weeks).

In our case, the VAD protocol resulted in a more rapid decrease in serum globulin than the daily dose of MP, alleviating the HVS-related systemic decline. However, this regimen failed to prevent disease progression as evidenced by paraplegia and urinary dysfunction due to spinal cord compression 24 days after VAD initiation. Furthermore, prolonged VAD use induces substantial bone marrow suppression and gastrointestinal toxicity, necessitating therapeutic modifications. A previous report described eight dogs with vertebral plasma cell tumors; three were diagnosed with MM. Two dogs survived for 17 and 26 months with chemotherapy alone (using the daily MP protocol, with weekly vincristine added in one case), whereas the other dog died acutely due to a vertebral compression fracture caused by osteolysis [15].

In humans, systemic chemotherapy alone is ineffective for acute spinal cord compression associated with MM. RT or surgical decompression is preferred unless only mild neurological deficits or epidural involvement are present [30]. In this case, palliative-intent RT was initiated promptly following neurological decline, and hind limb paralysis resolved within 2 days. Importantly, no evidence of recurrent spinal cord compression was seen on follow-up imaging before death, suggesting that timely local intervention played a critical role in neurological recovery in this canine vertebral MM case [30,31,32,33].

Due to the hematological and gastrointestinal toxicities observed during VAD and RT, the patient was administered a novel therapeutic agent, verdinexor, a selective inhibitor of nuclear export (SINE). Verdinexor (KPT-335) is an orally bioavailable inhibitor of exportin-1 (XPO1), which leads to nuclear retention of tumor suppressor proteins and subsequent apoptosis of malignant cells, and has been conditionally approved by the U.S. and FDA for the treatment of canine lymphomas [34,35].

Although their use in small animal oncology has been largely limited to canine lymphoma, SINEs have been investigated in human oncology for various hematologic and solid tumors, including non-Hodgkin lymphoma, relapsed/refractory MM, and soft tissue sarcomas [36,37,38,39]. To our knowledge, this is the first report to describe the clinical use of verdinexor in dogs with MM. Notably, in our case, verdinexor achieved the most stable and prolonged remission among all administered chemotherapeutic agents, with minimal toxicity.

However, the antitumor efficacy of verdinexor must be interpreted with caution. Delayed abscopal effects from prior RT or residual antitumor activity from the VAD protocol may have contributed to disease control [40,41]. Future studies are warranted to evaluate the use of SINE monotherapy before or in the absence of cytotoxic chemotherapy or RT for canine MM.

After disease relapse, combination protocols involving verdinexor, doxorubicin, melphalan, or bortezomib were explored. However, these combinations failed to elicit tumor regression and were associated with exacerbated adverse effects. Conversely, human studies have demonstrated that verdinexor synergizes with agents, such as melphalan, dexamethasone, bortezomib, and lenalidomide, potentially by re-sensitizing MM cells to previously ineffective therapies [42]. Future veterinary research is needed to determine optimal protocols that maximize efficacy and minimize toxicity when combined with other agents for refractory MM.

Another effective therapeutic approach involved combining a proteasome inhibitor with dexamethasone. In human MM, the combination of bortezomib, lenalidomide, and dexamethasone is currently the cornerstone regimen. Second-generation proteasome inhibitors such as ixazomib and carfilzomib have been used in bortezomib-refractory cases [43]. In veterinary oncology, the use of proteasome inhibitors has been reported in feline MM and experimental models of canine melanoma, osteosarcoma, lymphoma, and urothelial carcinoma [20,44,45,46].

In a feline MM case, a patient who failed MP therapy was successfully managed with bortezomib (0.7–1.0 mg/m^2^ SC twice weekly) and prednisolone (1 mg/kg PO q24h). Following 2 weeks of combination therapy, bortezomib monotherapy was continued for four cycles, achieving stable disease for approximately 160 days. Adverse events included grade 2 anorexia, grade 2 fatigue, grade 4 neutropenia, and grade 3 thrombocytopenia, with 0.7 mg/m^2^ suggested as a safe and effective feline dose [20].

In this case, bortezomib was initiated at 1.3 mg/m^2^ IV twice weekly, which is consistent with human starting doses [47]. Dexamethasone (1 mg/kg intravenously weekly) was administered concurrently. To determine the minimum effective dose, the bortezomib dose was escalated to 1.65 mg/m^2^ IV twice weekly. Tumor suppression, as measured by serum globulin reduction, was observed at doses ≥ 1.45 mg/m^2^. Attempts to combine low-dose bortezomib (0.7 mg/m^2^) with melphalan or verdinexor failed to maintain tumor control, indicating that lower doses may be insufficient for canine MM.

Despite its transient efficacy, the bortezomib–dexamethasone combination ultimately failed to provide long-term disease control. Moderate toxicities, such as grade 1 anemia and leukopenia, grade 2 thrombocytopenia, and grade 1 gastrointestinal signs (fever, anorexia, vomiting, diarrhea, and lethargy) were observed, particularly when combined with verdinexor. Second-line proteasome inhibitors, carfilzomib and ixazomib, showed limited or no antitumor effects in this patient.

Two potential factors contributed to this study’s limitations. First, the lack of access to immunomodulatory agents such as lenalidomide and thalidomide in veterinary practice hinders the use of full triple-drug regimens typically employed in human MM, where each agent contributes distinct yet complementary mechanisms of action [47,48]. Second, the limited number of published veterinary cases involving proteasome inhibitors makes it difficult to assess appropriate dosing and scheduling in dogs. Further clinical trials are required to refine the application of these agents in canine MM.

TKIs have been explored for the treatment of MM in humans; however, veterinary studies remain scarce. In one canine case, toceranib was administered to a patient with relapsed MM that was refractory to melphalan, prednisolone, and doxorubicin, achieving a 50% reduction in serum immunoglobulin A (IgA) levels and maintaining this response for 76 weeks [49]. However, in the present case, the addition of toceranib to bortezomib and dexamethasone failed to suppress tumor progression.

Conversely, sorafenib, a multi-kinase inhibitor targeting the RAF/MEK/ERK signaling pathway, proved to be more effective. This pathway is a key therapeutic target in human MM in which RAS/RAF mutations are common [50,51]. Although its role in canine MM remains unclear, upregulation of this pathway has been associated with shorter progression-free survival in canine lymphoma [52,53]. Therefore, sorafenib holds therapeutic promise for both lymphoma and MM. Sunitinib, a toceranib-related TKI, has shown limited efficacy in human MM and similarly failed in our canine patient [54].

However, the long-term use of sorafenib in our case led to significant dermatological and systemic side effects. Previous studies have reported toxicities at 30 mg/kg/day, including gastrointestinal, renal, adrenal, bone, and hematological complications, whereas doses of 10 mg/kg/day appear safer [55]. Another study using 5 mg/kg PO twice daily in dogs with hepatocellular carcinoma reported clinical efficacy; however, adverse effects such as grade 2 alopecia and grade 1 hyperpigmentation occurred in several dogs [56]. Our patient, treated with 3 mg/kg/day, developed alopecia, eruption, and hyperpigmentation in the ears, muzzle, neck, axilla, and limbs, along with hematuria and hematochezia, likely due to concurrent urinary tract infection and colonic bleeding. In humans, sorafenib induces cutaneous side effects in up to 10% of cases, commonly manifesting as painful palmoplantar plaques (“hand-foot syndrome”) [57]. A recent veterinary study suggested that weekly low-dose sorafenib (2–3 mg/kg) may reduce toxicity, although only 1 in 10 treated dogs demonstrated tumor regression [50]. Further dose-finding studies are essential to establish safe and effective sorafenib protocols for use in dogs.

In this case, treatment planning was primarily guided by the attending veterinarian’s clinical judgment, with the owners actively engaged in exploring therapeutic options beyond conventional veterinary protocols. This collaborative approach allowed for the consideration and use of human oncology agents that are costly and have limited veterinary precedent, including verdinexor, proteasome inhibitors (bortezomib, carfilzomib, ixazomib), and protein kinase inhibitors. The shift from standard chemotherapeutic regimens to targeted or novel agents was supported by the suboptimal or delayed control of hyperviscosity syndrome (HVS) with conventional protocols, the severe adverse effects observed during prolonged use, and the comparatively stable condition achieved with verdinexor and bortezomib. Although some therapies provided only transient benefit, this case illustrates how veterinary-led clinical decision-making, informed by owner engagement, can expand the therapeutic landscape for refractory MM while maintaining a focus on balancing efficacy, tolerability, and quality of life.

This case report presents a comprehensive overview of multimodal management strategies for relapsed and refractory MM in a dog. However, several limitations must be acknowledged. One major limitation was the inability to implement advanced therapies such as autologous stem cell transplantation (ASCT), immunotherapeutic agents, and plasmapheresis, which could potentially extend patient survival. In human medicine, ASCT remains the cornerstone of treatment for newly diagnosed MM after induction therapy with regimens including bortezomib, lenalidomide, and dexamethasone for 34 cycles [43,58]. Maintenance therapy with bortezomib- or lenalidomide-based regimens is administered after transplantation. These strategies have significantly improved complete response rates and progression-free survival. Moreover, ASCT is often considered during disease relapse.

Although ASCT has been explored in dogs with hematologic malignancies such as lymphoma and leukemia, and even in certain non-malignant hematologic disorders, to our knowledge, no published veterinary reports of its application in canine MM have been published [59,60,61,62,63,64,65]. Integration of stem cell-based therapies into veterinary MM is a promising area for future investigation.

In human medicine, the use of monoclonal antibodies such as daratumumab has revolutionized MM treatment by targeting CD38 on malignant plasma cells [43,66,67]. However, species specificity of these antibodies precludes their direct application in veterinary medicine. The development of canine-specific monoclonal antibodies is critically needed for both therapeutic purposes and for diagnostic applications, such as targeted PET-CT imaging to visualize MM lesions [68,69].

Another critical factor contributing to patient outcomes is the HVS, which ultimately plays a central role in mortality. HVS is an oncological emergency caused by increased levels of circulating cellular or proteinaceous elements, leading to impaired perfusion, organ damage, and hemorrhage, occurring in 20–40% of canine MM cases [5,6,7,8,11,70,71]. Treatment options include cytoreductive chemotherapy, hemodilution with intravenous fluids, phlebotomy, and plasmapheresis, with the latter being the most definitive intervention for acute decompensation.

In our case, emergency phlebotomy and red blood cell transfusion were attempted during the periods of severe hyperglobulinemia and anemia. However, without a rapid cytoreductive response to chemotherapy, these measures offer only transient benefits at a high cost and significant risks. Shortly before death, the patient developed urinary dysfunction without identifiable spinal cord compression on MRI, suggesting that HVS-induced neurotoxicity was responsible. At this point, plasmapheresis could have been life-saving. However, this procedure was not available in our clinical setting. The attempted transfusion, aimed at enabling further chemotherapy, exacerbated HVS and may have contributed to multi-organ failure. Plasmapheresis may have significantly prolonged survival had it been accessible.

Another limitation was the lack of definitive histopathological confirmation by biopsy or bone marrow examination. While the diagnosis was supported by classical findings—multiple osteolytic lesions, serum monoclonal gammopathy, Bence–Jones proteinuria, and cytological infiltration of plasma cells in the spleen and liver—tissue-based confirmation would have increased the diagnostic certainty [5,7,11].

Moreover, treatment response and relapse assessment relied on serum globulin concentrations and imaging findings. Currently, no standardized protocol exists for monitoring treatment response in canine MM [9]. A recent study suggested that densitometric monitoring of M-protein provided a better correlation with clinical events and survival outcomes than traditional serum globulin measurements [72]. The application of these methods may yield more precise and predictive insights.

## 4. Conclusions

In summary, this case highlights both the therapeutic potential and the clinical challenges of managing relapsed, refractory MM in dogs. While classical regimens such as MP and VAD remain important, treatment selection in this case evolved based on the patient’s clinical course, therapeutic response, and tolerability to prior protocols. This approach facilitated the integration of novel therapies—including SINE, proteasome inhibitors, and TKIs—beyond conventional veterinary protocols. Verdinexor achieved the most prolonged remission with minimal toxicity, bortezomib demonstrated efficacy within a defined therapeutic window, and sorafenib showed potential as a kinase-targeted therapy. However, the absence of advanced modalities such as ASCT, monoclonal antibodies, and plasmapheresis significantly limited long-term disease control. Collaborative efforts in veterinary oncology are essential to establish evidence-based protocols for advanced diagnostics and novel therapeutics, enabling optimized, individualized treatment strategies for canine MM.

## Figures and Tables

**Figure 1 animals-15-02405-f001:**
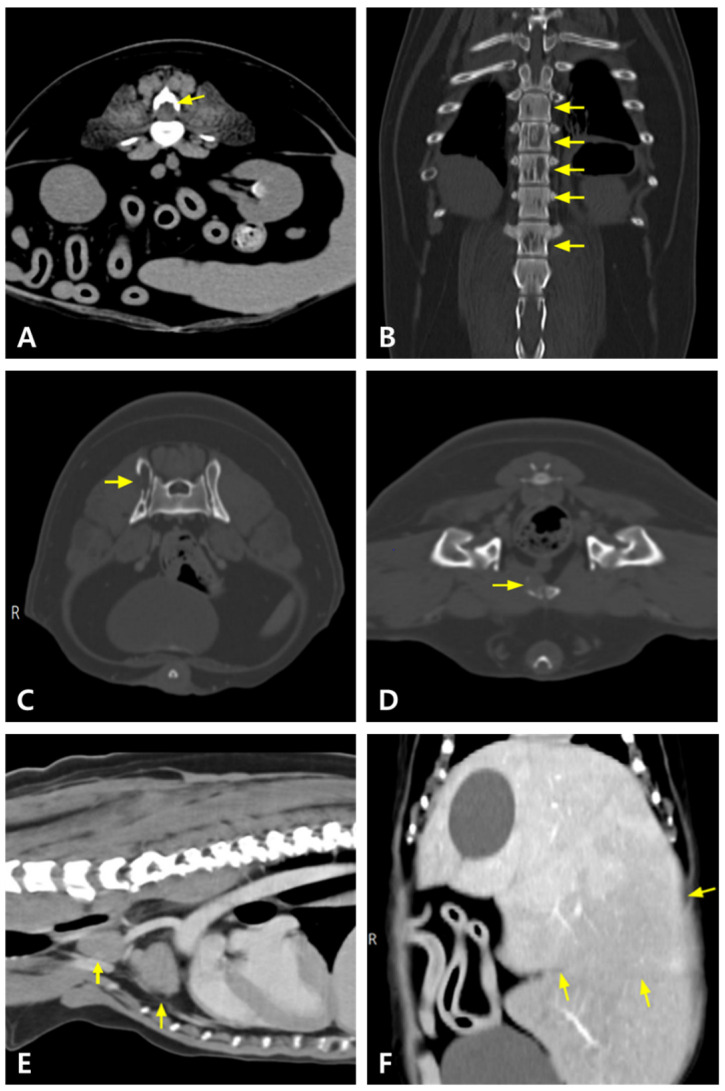
Computed tomography (CT) findings in this dog. The mass (arrow) at the level of the third lumbar vertebra (L3) and vertebral osteolysis are identified on the transverse view (pre-contrast) (**A**). Regions of decreased attenuation are observed in multiple areas of the thoracic and lumbar spine (pre-contrast) (arrows) (**B**). Osteolytic lesions are noted in the iliac (arrow) (**C**) and pubic (arrow) (**D**) regions (post-contrast). Mediastinal lymph nodes (arrows) (**E**) and liver (arrows) (**F**) are markedly enlarged (post-contrast).

**Figure 2 animals-15-02405-f002:**
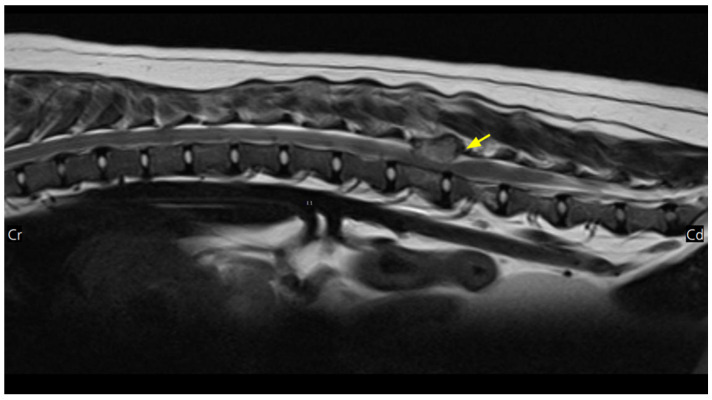
Magnetic resonance imaging (MRI) findings in this dog. A mass (arrow) at the L3–L4 level is compressing the spinal cord on the T2-weighted, pre-contrast sagittal view.

**Figure 3 animals-15-02405-f003:**
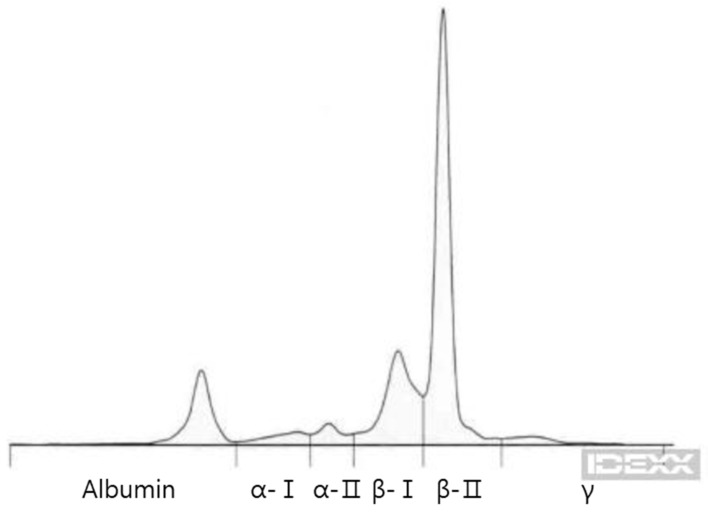
Electrophoretogram of the dog. A tall and narrow peak in the β-II region.

**Figure 4 animals-15-02405-f004:**
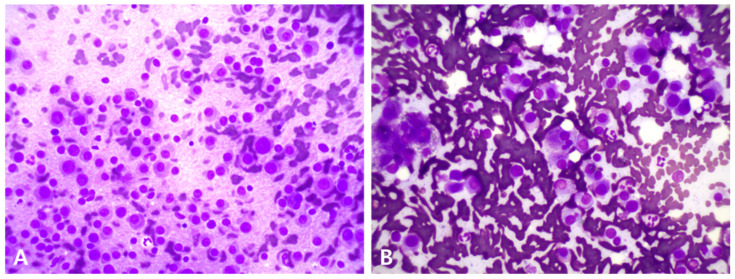
Cytology of the spleen (**A**) and liver (**B**). Round cells are seen predominantly with blue cytoplasm, round nuclei, and perinuclear clear zone (500× magnification, Diff-Quik stain).

**Figure 5 animals-15-02405-f005:**
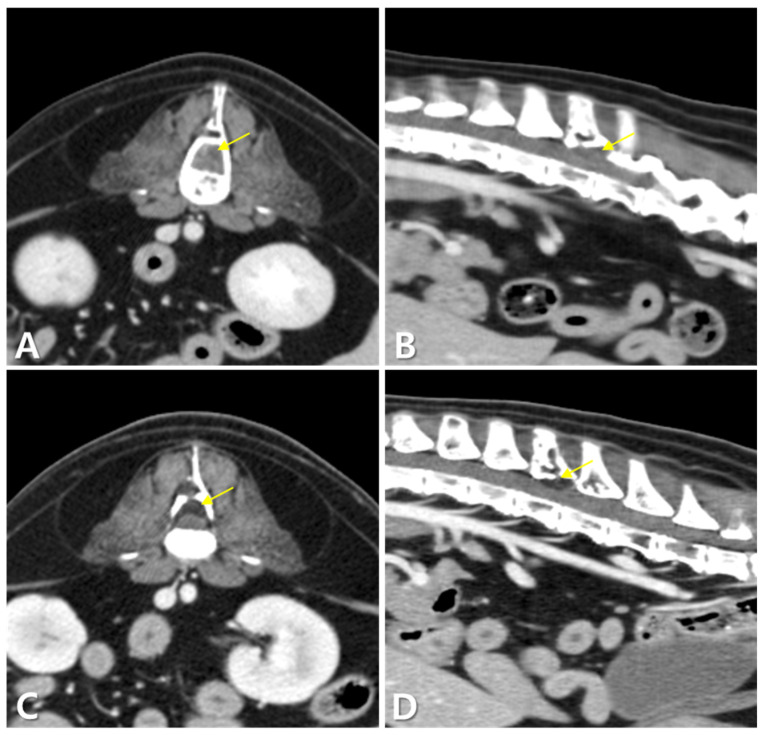
Post-contrast CT images of the mass at the L3–L4 vertebral level before (**A**,**B**) and after (**C**,**D**) radiation therapy. (**A**,**B**) Pre-radiation therapy images show a soft tissue attenuation mass (arrows in (**A**,**B**)) with dorsal compression of the spinal cord. (**C**,**D**) Post-radiation therapy images reveal that the mass (arrows in (**C**,**D**)) has transformed to fat attenuation with significant reduction in size and alleviated compression.

**Figure 6 animals-15-02405-f006:**
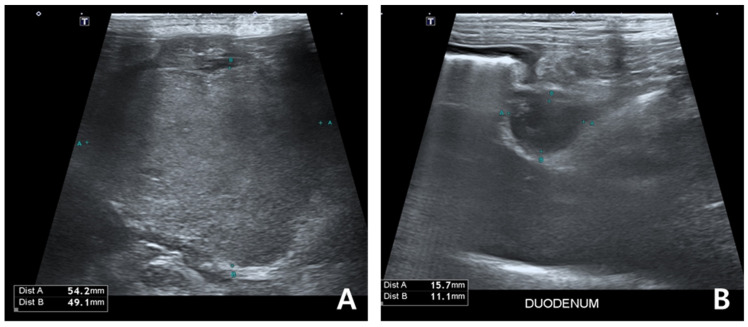
Abdominal ultrasonography on day 189 in the dog. A mass measuring approximately 5 cm in diameter is observed in the abdominal cavity, suspected to be an enlarged hepatic lymph node (**A**). An additional mass is identified at the level of the duodenum (**B**).

**Figure 7 animals-15-02405-f007:**
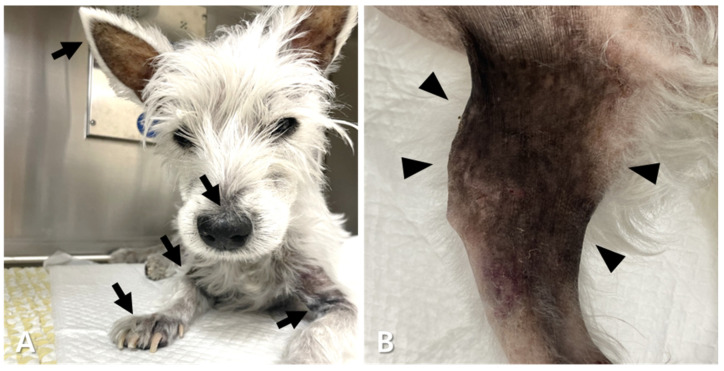
Alopecia, skin eruption, and hyperpigmentation are observed on the pinna, muzzle, neck, axillary region, and forelimbs ((**A**); arrows), and on the inguinal region and hind limbs ((**B**); arrow heads).

**Figure 8 animals-15-02405-f008:**
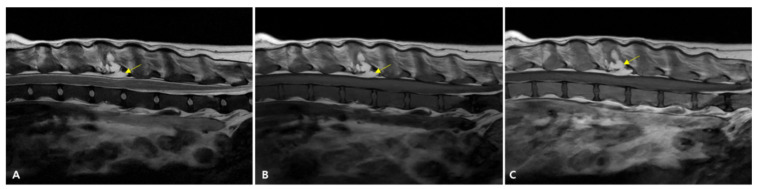
MRI images of the L3–L4 vertebral level on day 395. (**A**) T2-weighted image (T2WI), (**B**) T1-weighted image (T1WI), and (**C**) post-contrast T1-weighted image. The treated L3 mass (arrows in (**A**–**C**)) shows high signal intensity on both T2WI and T1WI, consistent with fat, and has decreased in size. Spinal cord compression is nearly resolved.

**Figure 9 animals-15-02405-f009:**
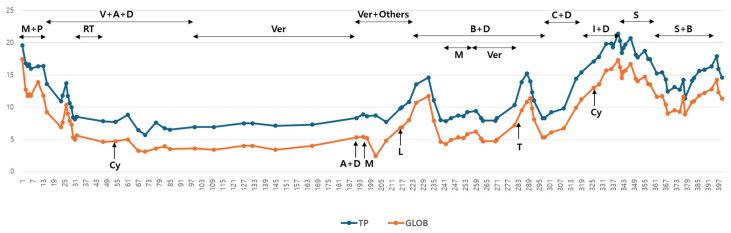
Anti-cancer treatment modalities and corresponding serum total protein (TP) and globulin (GLOB) concentrations. M: Melphalan; P: Prednisolone; V: Vincristine; A: Doxorubicin; D: Dexamethasone; RT: Radiation Therapy; Ver: Verdinexor; B: Bortezomib; C: Carfilzomib; I: Ixazomib; S: Sorafenib; Cy: Cyclophosphamide; L: Lomustine; T: Toceranib.

**Table 1 animals-15-02405-t001:** Hematological parameters of the dog.

	Day 0	Day 9	Day 81	Day 165	Day 190	Day 238	Day 289	Day 315	Day 361	Day 392	Reference Range
Hematocrit (%)	30.8	30.1	19	41.4	38.6	41.4	36.1	28.2	26.5	19.8	37.3–61.7
White Blood Cells (k/μL)	6.46	5.6	8.88	6.88	11.29	6.75	3.0	7.89	6.59	14.11	5.05–16.76
Platelet (k/μL)	197	215	54	193	84	90	55	121	26	22	148–484
Total protein (g/dL)	19.6	16.3	6.5	7.3	8.3	8.0	14.0	14.4	15.2	17.9	5.2–8.2
Albumin (g/dL)	2.2	2.4	3.0	3.3	3	3.3	2.6	4.5	3.6	3.6	2.3–4.0
Globulin (g/dL)	17.4	13.9	3.5	4	5.3	4.6	11.4	9.9	11.6	14.2	2.5–4.5
ALT (U/L)	540	309	14	38	41	59	>1000	-	94	649	10–125
ALP (U/L)	277	495	534	280	563	264	836	807	544	436	23–212
BUN (mg/dL)	11	27	29.5	24	11	21	-	45	42	55	7–27
Creatinine (mg/dL)	1.0	1.8	0.7	1.4	0.6	0.8	-	1.7	2.3	2.4	0.51.8
Calcium (μg/dL)	12.9	12.2	10	9.9	9.8	-	-	11.7	-	12.3	7.9–12.0
Phosphate (mg/mL)	5.5	7.2	4.1	5.3	3.6	6.1	-	7.3	5.5	12.5	2.5–6.8

ALT: Alanine Aminotransferase; ALP: Alkaline Phosphatase; BUN: Blood Urea Nitrogen. Day 0: Baseline, prior to initiation of any treatment. Day 9: 9 days after initiation of melphalan + prednisolone combination therapy. Day 81: 73 days after initiation of vincristine + doxorubicin + dexamethasone combination protocol. Day 165: 58 days after initiation of verdinexor. Day 190: 83 days after initiation of verdinexor. Day 238: 14 days after initiation of bortezomib + dexamethasone combination therapy. Day 289: 68 days after initiation of bortezomib + dexamethasone combination therapy. Day 315: 20 days after initiation of carfilzomib + dexamethasone combination therapy. Day 360: 20 days after initiation of sorafenib. Day 392: 33 days after initiation of bortezomib + sorafenib combination therapy.

**Table 2 animals-15-02405-t002:** Overall treatment summary for the patient.

Survival time after initial treatment for multiple myeloma	398 days
Side effects of VAD	(VCOG-CTCAE v2) Grade 3 anemia, grade 2 thrombocytopenia, and grade 1 anorexia, vomiting, altered appetite, and lethargy
Side effects of verdinexor	None observed
Side effects of verdinexor + doxorubicin + melphalan + dexamethasone	(VCOG-CTCAE v2) Grade 2 anemia, grade 3 leukopenia, grade 4 thrombocytopenia, and grade 1 abdominal distention, anorexia, vomiting, diarrhea, and lethargy
Side effects of bortezomib + dexamethasone	(VCOG-CTCAE v2) Grade 2 thrombocytopenia and grade 1 diarrhea
Side effects of bortezomib + dexamethasone + melphalan	Grade 1 diarrhea and grade 1 keratitis
Side effects of bortezomib + dexamethasone + verdinexor	(VCOG-CTCAE v2) Grade 1 anemia and leukopenia, grade 2 thrombocytopenia, and grade 1 fever, anorexia, vomiting, diarrhea, and lethargy
Side effects of carfilzomib + dexamethasone	Not remarkable observation (clinical signs presumed to be tumor-related)
Side effects of ixazomib + dexamethasone	Not remarkable observation (anemia presumed to be tumor-related)
Side effects of bortezomib + sorafenib	(VCOG-CTCAE v2) Grade 2 anemia and thrombocytopenia, grade 2 alopecia and hyperpigmentation, grade 1 skin ulceration, grade 2 colitis, and grade 2 urinary tract infection

VCOG-CTCAE v2: Veterinary Cooperative Oncology Group-Common Terminology Criteria for Adverse Events [27].

## Data Availability

The original contributions presented in this study are included in the article. Further inquiries can be directed to the corresponding authors.

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
