# Peer review of "Refractory Multiple Myeloma in a West Highland White Terrier: Clinical Presentations and Therapeutic Interventions"

_animals, 2025, doi:10.3390/ani15162405_

Round 1

Reviewer 1 Report

Comments and Suggestions for Authors

I carefully read the manuscript titled ‘Refractory Multiple Myeloma in a West Highland White Terrier: Clinical Presentations, Imaging Findings, and Therapeutic Interventions’. This study provides a detailed examination of the multimodal management strategies employed for treating relapsed and refractory multiple myeloma (MM) in a dog. The introduction provides a comprehensive overview, detailing all therapeutic approaches with numerous bibliographic references. The presentation of the case's progression is analytical. However, while the title includes "imaging findings," there is a lack of description regarding the imaging diagnostic approach in both the introduction and, more significantly, in the discussion. Additionally, corresponding references related to the diagnostic approach are absent. Therefore, it is recommended that more information be included, or alternatively, the term "imaging findings" should be removed from the title. A few comments and remarks follow below.

Line 42: The Keyword multiple myeloma is also mentioned in the title; no need to repeat, it could be replaced with another word.

Line 53, 62: Please add a reference.

Line 194: Where is Figure 5?

Line 204-206: The performance of X-rays and ultrasound is noted for the first time; however, if I understand correctly, this refers to a follow-up. If these imaging methods were conducted during the diagnosis, they should be mentioned earlier in the text. If they were not, please clarify when they were performed and their purpose.

Line 247-248: ‘Concurrent urinary tract infection resulted in hematuria and hematochezia.’ Is this correct? The UTI resulted in hematochezia?

Line 259: Please comment on the arrows in the images.

Author Response

Revision Response Letter

Manuscript ID: animals-3786318
Title: Refractory Multiple Myeloma in a West Highland White Terrier: Clinical Presentations and Therapeutic Interventions

Dear Editors and Reviewers,

We sincerely thank the reviewers for their valuable and constructive feedback on our manuscript. We have carefully revised the text to address each comment and believe these changes have strengthened the clarity, completeness, and scientific value of our report. Our detailed point-by-point responses are provided below

Reviewer 1

  1. I carefully read the manuscript titled ‘Refractory Multiple Myeloma in a West Highland White Terrier: Clinical Presentations, Imaging Findings, and Therapeutic Interventions’. This study provides a detailed examination of the multimodal management strategies employed for treating relapsed and refractory multiple myeloma (MM) in a dog. The introduction provides a comprehensive overview, detailing all therapeutic approaches with numerous bibliographic references. The presentation of the case's progression is analytical. However, while the title includes "imaging findings," there is a lack of description regarding the imaging diagnostic approach in both the introduction and, more significantly, in the discussion. Additionally, corresponding references related to the diagnostic approach are absent. Therefore, it is recommended that more information be included, or alternatively, the term "imaging findings" should be removed from the title. A few comments and remarks follow below.

Answer:

Thank you for your valuable comment regarding the title. Although CT, MRI, ultrasonography, and radiography were consistently performed during the diagnostic process, as you pointed out, this manuscript focuses more on the treatment and its clinical course. Therefore, we have revised the title to: “Refractory Multiple Myeloma in a West Highland White Terrier: Clinical Presentations and Therapeutic Interventions.”

  1. Line 42: The Keyword multiple myeloma is also mentioned in the title; no need to repeat, it could be replaced with another word.

Answer:

Thank you for reminding us of the MDPI guidelines regarding keywords. We have removed the term “multiple myeloma” from the keywords, as it is already included in the title.

  1. Line 53, 62: Please add a reference.

Answer:

We have added a reference and slightly refined the sentence at Line 53 for greater clarity. Additionally, we have added a reference at Line 62.

  1. Line 194: Where is Figure 5?

Answer: Figure 5, which was missing in the original submission, has been added in the revised manuscript.

  1. Line 204-206: The performance of X-rays and ultrasound is noted for the first time; however, if I understand correctly, this refers to a follow-up. If these imaging methods were conducted during the diagnosis, they should be mentioned earlier in the text. If they were not, please clarify when they were performed and their purpose

Answer:

As you correctly pointed out, the patient underwent follow-up radiographic and ultrasonographic examinations at least once a month, and in some periods, up to once a week. The finding described at Line 204 refers to a new abnormality that was not detected in previous examinations. To emphasize this, we have added the following sentence at Line 144: “Along with chemotherapy, periodic blood tests, thoracic and abdominal radiographs, and abdominal ultrasonography were performed to monitor the progression and characteristics of the lesions over time.”

  1. Line 247-248: ‘Concurrent urinary tract infection resulted in hematuria and hematochezia.’ Is this correct? The UTI resulted in hematochezia?

Answer: We apologize for the awkward phrasing in the original text. The sentence has been revised to: “Urinalysis revealed hematuria associated with concurrent urinary tract infection, and hematochezia due to colitis was also observed.”

  1. Line 259: Please comment on the arrows in the images

Answer:

As advised, arrows indicating the lesions have been added to Figure 7.

Reviewer 2 Report

Comments and Suggestions for Authors

Dear authors, thank you for your work. It's an interesting article, which reveals a very well-worked clinical case. The biggest problem is the failure to confirm the diagnosis by histopathology. I don't think there is any other possible diagnosis, but since the animal was anaesthetised or sedated for imaging tests (I presume), it is regrettable that a biopsy was not carried out.

I have some comments and doubts about your work, which I'll explain below

Were the citologies stained with H&E? It´s unusual

It also becomes confusing to understand the sequence of drugs administered and when they were stopped. The initial therapy with Melphalan (0.1 mg/kg PO q24h) and prednisolone (0.5 mg/kg PO q24h) was discontinued when? or prednisolone was maintained throughout the treatment? a schematic representation of the drug sequence would be interesting. this information appears much later, in figure 9, but it would be good to make it available as soon as the drug is mentioned

was the animal hospitalised during treatment? were the owners given instructions on how to care for faeces and urine after chemotherapy?

I have some ethical questions about the use of unlicensed pharmaceuticals in dogs. Was there informed consent from the owners? Was the animal's welfare taken care of? Were pain and other side effects controlled? Are the drugs in question licensed for use in veterinary medicine in the country where the clinical case was monitored? in line 459 the authors refer "emergency phlebotomy and red blood cell transfusion" . This is the first time it has been mentioned. When it was done?

what was the rationale behind the sequence of drugs? what were the reasons for the first choice and afterwards?

Reading your article gives me the impression that the animal is a secondary element in this whole process. It would be good if the supporting medical care, which has certainly been done, were added to the text.

Thank you agin for your work, I think it will be very valuable for clinical practice

Author Response

Revision Response Letter

Manuscript ID: animals-3786318
Title: Refractory Multiple Myeloma in a West Highland White Terrier: Clinical Presentations and Therapeutic Interventions

Dear Editors and Reviewers,

We sincerely thank the reviewers for their valuable and constructive feedback on our manuscript. We have carefully revised the text to address each comment and believe these changes have strengthened the clarity, completeness, and scientific value of our report. Our detailed point-by-point responses are provided below

Reviewer 2

  1. The biggest problem is the failure to confirm the diagnosis by histopathology

Answer: Thank you for this important comment. We also consider the lack of histopathological confirmation to be the most regrettable limitation of this case. At the initial presentation, the main lesion was located in the vertebrae, and the owners declined a biopsy due to the associated risks. As other diagnostic criteria strongly suggested multiple myeloma, we proceeded with a presumptive diagnosis and initiated treatment. In future similar cases, we will make a stronger effort to emphasize to owners the importance of histopathological confirmation, particularly bone marrow biopsy, for definitive diagnosis.

  1. Were the cytologies stained with H&E? It´s unusual

Answer:

Thank you for pointing this out. This was an error in the original text — the cytology was performed in-house using Diff-Quik stain. We have corrected this in the text and updated the description in Figure 4 accordingly.

  1. It also becomes confusing to understand the sequence of drugs administered and when they were stopped.

Answer:

We agree with your observation that, although the sequence of numerous drugs administered and discontinued over the long treatment period was summarized in Figure 9, the description in the text was not sufficiently clear. We have revised the Case Description to provide a clearer narrative of the treatment timeline and referenced Figure 9 at the first mention of chemotherapy for easier reader comprehension.

Line 166~ 168: Consequently, melphalan and prednisolone were discontinued, and a vincristine-doxorubicin-dexamethasone (VAD) chemotherapy protocol was initiated to induce a rapid reduction in blood protein concentration

Line 224: On day 224, verdinexor was discontinued, and bortezomib (1.3 mg/m² IV twice weekly) and dexamethasone (1 mg/kg IV weekly) were administered.

Line 242: Toceranib (3 mg/kg PO, thrice weekly) was added, but no clinical improvement was observed.

  1. Was the animal hospitalised during treatment? were the owners given instructions on how to care for faeces and urine after chemotherapy?

Answer:

During the treatment period, the patient was managed on an outpatient basis except for hospitalization during radiation therapy, placement of a urinary catheter due to urinary dysfunction, detection of anemia requiring intensive monitoring, and administration of blood transfusions. The owners were instructed to wear gloves when handling the dog’s urine, feces, or vomitus. There were no young children, elderly individuals, or immunocompromised persons in the household.

  1. I have some ethical questions about the use of unlicensed pharmaceuticals in dogs. Was there informed consent from the owners?

Answer:

Thank you for your concern. The treatment decisions for this patient were made with significant input from the owners, including the use of drugs not previously applied in veterinary medicine. In fact, some of the medications—such as carfilzomib and ixazomib—are extremely costly, and their use would not have been possible without the owners’ strong interest and willingness to proceed. After experiencing severe adverse effects with the VAD protocol, the owners became hesitant about conventional chemotherapeutic agents. The prolonged disease control with verdinexor without major toxicity, the limited efficacy and notable side effects of conventional chemotherapy upon relapse, and the relatively stable condition achieved with bortezomib all contributed to the owners’ interest in exploring novel human oncology drugs. The owners were also highly supportive of preparing this manuscript, as they wished to preserve their dog’s memory through the publication of this case.

We have incorporated this content into the Discussion section (line 443) as follows:

Line 443 In this case, treatment planning was primarily guided by the attending veterinarian’s clinical judgment, with the owners actively engaged in exploring therapeutic options beyond conventional veterinary protocols. This collaborative approach allowed the consideration and use of human oncology agents that are costly and have limited veterinary precedent, including verdinexor, proteasome inhibitors (bortezomib, carfilzomib, ixazomib), and protein kinase inhibitors. The shift from standard chemotherapeutic regimens to targeted or novel agents was supported by the suboptimal or delayed control of hyperviscosity syndrome (HVS) with conventional protocols, the severe adverse effects observed during prolonged use, and the comparatively stable condition achieved with verdinexor and bortezomib. Although some therapies provided only transient benefit, this case illustrates how veterinary-led clinical decision-making, informed by owner engagement, can expand the therapeutic landscape for refractory MM while maintaining a focus on balancing efficacy, tolerability, and quality of life.

  1. Were pain and other side effects controlled?

Answer:

All cancer- and chemotherapy-related adverse effects mentioned in the manuscript were actively managed. Over the course of approximately one year, the owners and the medical team tried every possible measure, including multiple antiemetics, analgesics, fluid therapy, nutritional supplements, a home-cooked diet, and even oriental physical therapy. As the focus of this manuscript is primarily on anticancer treatment, these supportive measures were not described in detail in order to maintain the main scope of the report.

  1. Are the drugs in question licensed for use in veterinary medicine in the country where the clinical case was monitored?

Answer: Regarding your concern, we would like to clarify the relevant regulations on the use of human pharmaceuticals in veterinary medicine in Korea:
a. The use of human medicines for animals is, in principle, permitted under Korean law. When intended for the treatment of animals, veterinarians are legally allowed to purchase and administer human prescription drugs, as stipulated in the Pharmaceutical Affairs Act, the Veterinary License Act, and the Regulation on the Control of Veterinary Drugs (Pharmaceutical Affairs Act Article 2; Regulation on the Control of Veterinary Drugs Article 22).
b. Such use is allowed only when there is no veterinary-specific drug available or when no effective treatment can be expected from available veterinary drugs. This policy aligns with international standards applied in countries such as the United States, the United Kingdom, and Japan.

  1. line 459 the authors refer "emergency phlebotomy and red blood cell transfusion". This is the first time it has been mentioned. When it was done?

Answer: Phlebotomy and blood transfusion were attempted only once on Day 336, concurrently with transfusion, but the procedure did not result in a marked reduction in proteinemia, and levels returned to baseline within two days. Given the lack of positive impact relative to the cost and patient discomfort, the following sentence was added to Line 251: “Emergency phlebotomy and red blood cell transfusion were attempted but proved ineffective.”

  1. what was the rationale behind the sequence of drugs? what were the reasons for the first choice and afterwards?

Answer: As is well known, melphalan combined with prednisolone is the first-choice protocol for treating multiple myeloma (MM), and we initially adopted this regimen. However, the effect on hyperviscosity syndrome (HVS) was slow, during which time HVS-related complications, such as pancreatitis, developed. Consequently, we initiated the VAD protocol, which is recognized in veterinary medicine as a rescue regimen. This resulted in a dramatic reduction in HVS; however, the patient developed urinary dysfunction due to mechanical spinal cord compression from the vertebral tumor. We therefore combined radiation therapy (RT) with VAD. After several additional VAD cycles, although the patient’s condition remained stable, the chemotherapy-induced adverse effects were significant, prompting us to discontinue conventional chemotherapy and try verdinexor, a novel anticancer agent. Verdinexor allowed the most stable long-term disease control in this case. Upon relapse of MM, we administered a short course of conventional chemotherapy with some success; however, given the previous severe adverse effects associated with long-term conventional chemotherapy, the owners were reluctant to continue. They had researched bortezomib, a proteasome inhibitor commonly used in human oncology and previously reported in a feline MM case, and requested its use. Although the duration of tumor control with bortezomib was not prolonged, it was effective in stabilizing the disease, and the owners were satisfied as the drug’s adverse effects were not severe (except when combined with conventional chemotherapy, which increased toxicity).

For the owners, the highest priority was that their dog maintained a good appetite. Therefore, rather than continuing conventional chemotherapy, which significantly reduced the patient’s general condition, they preferred to try targeted therapies for MM—even if costly and not previously reported in dogs. Carfilzomib and ixazomib, as next-generation proteasome inhibitors, were also requested by the owners after reviewing human medical literature, despite unknown effective doses in dogs and high cost. Unfortunately, these did not produce the expected results, so the focus shifted to protein kinase inhibitors (PKIs). These were tried first as monotherapy, and later in combination with bortezomib, as in human protocols. Initial responses were encouraging, but the patient’s overall condition deteriorated severely during treatment, leading to the decision to discontinue further chemotherapy. We acknowledge that the process of trying these newer treatment approaches may appear unconventional; however, both the owners and the medical team prioritized not simply tumor reduction, but the patient’s overall well-being—ensuring minimal discomfort, maintaining appetite, and preserving normal urination and defecation—when selecting each treatment.

  Line 443 In this case, treatment planning was primarily guided by the attending veterinarian’s clinical judgment, with the owners actively engaged in exploring therapeutic options beyond conventional veterinary protocols. This collaborative approach allowed the consideration and use of human oncology agents that are costly and have limited veterinary precedent, including verdinexor, proteasome inhibitors (bortezomib, carfilzomib, ixazomib), and protein kinase inhibitors. The shift from standard chemotherapeutic regimens to targeted or novel agents was supported by the suboptimal or delayed control of hyperviscosity syndrome (HVS) with conventional protocols, the severe adverse effects observed during prolonged use, and the comparatively stable condition achieved with verdinexor and bortezomib. Although some therapies provided only transient benefit, this case illustrates how veterinary-led clinical decision-making, informed by owner engagement, can expand the therapeutic landscape for refractory MM while maintaining a focus on balancing efficacy, tolerability, and quality of life.

  1. Reading your article gives me the impression that the animal is a secondary element in this whole process. It would be good if the supporting medical care, which has certainly been done, were added to the text.

Answer:

As mentioned in our response to Comment 6, numerous additional supportive measures were attempted. To address this in the manuscript, we have added the following sentence at Line 295: “During an approximate one-year treatment period, the patient received both inpatient and outpatient care, alongside a variety of supportive interventions aimed at managing tumor-related clinical signs and mitigating chemotherapy-associated adverse effects. These included supportive nutritional therapy, analgesics, antioxidants, antiemetics, antidiarrheals, appetite stimulants, and intravenous fluid administration.”
